# How to Engage Health Care Workers in the Evaluation of Hospitals: Development and Validation of BSC-HCW1—A Cross-Sectional Study

**DOI:** 10.3390/ijerph19159096

**Published:** 2022-07-26

**Authors:** Faten Amer, Sahar Hammoud, Haitham Khatatbeh, Huda Alfatafta, Abdulsalam Alkaiyat, Abdulnaser Ibrahim Nour, Dóra Endrei, Imre Boncz

**Affiliations:** 1Doctoral School of Health Sciences, Faculty of Health Sciences, University of Pécs, 7621 Pécs, Hungary; hammoud.sahar@etk.pte.hu (S.H.); khatatbeh.haitham@etk.pte.hu (H.K.); huda.alfatafta@etk.pte.hu (H.A.); 2Institute for Health Insurance, Faculty of Health Sciences, University of Pécs, 7621 Pécs, Hungary; endrei.dora@pte.hu (D.E.); imre.boncz@etk.pte.hu (I.B.); 3Division of Public Health, Faculty of Medicine and Health Sciences, An-Najah National University, Nablus P.O. Box 7, Palestine; abdulsalam.alkaiyat@unibas.ch; 4Faculty of Economics and Social Sciences, An-Najah National University, Nablus P.O. Box 7, Palestine; a.nour@najah.edu; 5National Laboratory for Human Reproduction, University of Pécs, 7624 Pécs, Hungary

**Keywords:** balanced scorecard, health personnel, satisfaction, loyalty, hospital, performance evaluation, quality

## Abstract

Organizations worldwide utilize the balanced scorecard (BSC) for their performance evaluation (PE). This research aims to provide a tool that engages health care workers (HCWs) in BSC implementation (BSC-HCW1). Additionally, it seeks to translate and validate it at Palestinian hospitals. In a cross-sectional study, 454 questionnaires were retrieved from 14 hospitals. The composite reliability (CR), interitem correlation (IIC), and corrected item total correlation (CITC) were evaluated. Exploratory factor analysis (EFA) and confirmatory factor analysis (CFA) were used. In both EFA and CFA, the scale demonstrated a good level of model fit. All the items had loadings greater than 0.50. All factors passed the discriminant validity. Although certain factors’ convergent validity was less than 0.50, their CR, IIC, and CITC were adequate. The final best fit model had nine factors and 28 items in CFA. The BSC-HCW1 is the first self-administered questionnaire to engage HCWs in assessing the BSC dimensions following all applicable rules and regulations. The findings revealed that this instrument’s psychometric characteristics were adequate. Therefore, the BSC-HCW1 can be utilized to evaluate BSC perspectives and dimensions. It will help managers highlight which BSC dimension predicts HCW satisfaction and loyalty and examine differences depending on HCWs’ and hospital characteristics.

## 1. Introduction

### 1.1. History of Balanced Scorecard (BSC)

The BSC was first suggested by Norton and Kaplan in 1992, and it included four perspectives: financial, customer, internal process, and knowledge and growth [1]. The last perspective was also referred to as the learning, development, or innovation perspective in certain earlier iterations of the BSC [2]. The customer perspective included focused mainly on the patients. However, in some implementations, it also included health care workers (HCWs), or both [2,3].

BSC perspective assessment provides managers with a comprehensive performance evaluation (PE) approach [2]. The original BSC generation simply included the four perspectives influenced by the organizational strategy. The first generation of BSCs is shown in Figure 1. Researchers in the second generation suggested the presence of causal linkages between the key performance indicators (KPIs) of the BSC perspectives [4], as seen in Figure 2. Strategic map was titled for these causal relationships [2,3]. Studies utilized strategic maps in the PE based on extracting data mainly from hospital records to evaluate all the perspectives [2]. In a few cases, stakeholders; patients or HCWs or both, were involved in BSC implementations only to evaluate their satisfaction [2]. However, it was found that there was a lack of stakeholder engagement in the evaluation of the rest of the perspectives and dimensions in BSC implementations [2,3,5]. Moreover, none of the BSC implementations analyzed the strategic maps based on the evaluation of the BSC perspectives from stakeholders’ opinions and observations. The third generation of BSC included goals and action plans for each KPI. Sustainability, which was sometimes referred to as the external, social and environmental perspective [2], was eventually incorporated as the fifth pillar of BSC [6].

### 1.2. The Impact of BSC

In our first BSC systematic review [3], BSC deployment has been shown to enhance health care organization’s (HCO’s) financial performance [3]. Furthermore, BSC proved to be effective in increasing patient satisfaction. However, it did not prove effective in enhancing the satisfaction rate of HCWs [3]. This was due to many reasons. First, most of the implementations focused on measuring HCW satisfaction as a sole indicator [2,3]. Second, although strategic maps were utilized based on hospital record data in BSC implementations, there has been a lack of analysis of the factors that impact or predict HCW satisfaction based on HCW opinions and observations [3]. Third, despite the researchers have pointed to the importance of patient and HCW engagement in the process of PE and delivery improvement [8,9,10], the reviews [2,5] revealed that there had been a lack of engaging stakeholders in BSC implementations, such as engaging patients and HCWs. Based on the review [3], we came to a recommendation that engaging HCWs in BSC implementations might provide a solution to the issue of stagnant levels of satisfaction among HCWs in BSC implementations. In addition, the participation of HCWs will aid HCO managers and researchers in their efforts to obtain a better grasp of the BSC strategic maps as well as the causal relationships between KPIs based on the perspectives of HCWs. Moreover, we think that the participation of HCWs in BSC implementations will result in an even greater improvement in both the financial performance and the level of satisfaction perceived by patients.

### 1.3. BSC Perspectives and Dimensions

In our second systematic review of BSC [2], we identified the perspectives and dimensions that were the most important and most frequently used in BSC implementations in the health care sector. Figure 3 represents a summary of the perspectives, major dimensions, and subdimensions that were more frequently used and deemed essential by health care managers worldwide. We also found that in a manner similar to the inadequate emphasis placed on HCW satisfaction during BSC implementations in HCOs, the notion of HCW loyalty was almost never taken into consideration [2,3]. A review in the business field [11] found a strong positive relationship between satisfaction and loyalty. However, these relationships were found to be moderated by different factors, such as demographics and setting type. We think that understanding HCWs’ loyalty attitudes may assist hospital managers in expecting HCWs’ future behavior. This will provide insight to managers when evaluating their hospitals’ performance, building their plans, and allocating their resources.

### 1.4. Health Care Sector and HCWs in Palestine

The health care system in Palestine is described to be fragile and incoherent [12,13]. This was referred to as the political and economic situation [14]. Furthermore, the doctor–patient miscommunication in Palestine [15,16], as well as the coronavirus-19 (COVID-19) pandemic, added an extra layer of challenge to the Palestinian health care sector [17,18,19]. These challenges exploited obstacles to improving the performance of HCOs in Palestine and highlighted the importance of involving HCWs in the process of improving the performance of HCOs through a better understanding of Palestinian HCW perceptions. A BSC implementation [20] found that there are limited validated instruments to measure management practices in low- and middle-income countries (LMICs), and all are unrelated to BSC. There is also limited research on PE for hospitals in Palestine. To our knowledge, there is no validated instrument to evaluate the performance of Palestinian hospitals to date.

In this research, the first aim is to develop an instrument that engages HCWs in a comprehensive assessment of BSC perspectives and dimensions based on HCWs needs (BSC-HCW1). The second aim of this research is to customize the developed instrument at Palestinian hospitals, translate it into Arabic, and validate it.

### 1.5. The Conceptual Framework

The dimensions and KPIs that emerged from our BSC systematic review [2] served as the basis for our conceptual model development. Because of the large number of KPIs, we narrowed our focus to those that are directly related to the demands of HCWs at each BSC perspective. In tandem with reviewing 34 studies in the literature [2,6,21,22,23,24,25,26,27,28,29,30,31,32,33,34,35,36,37,38,39,40,41,42,43,44,45,46,47,48,49,50,51,52], we separately examined 77 causal linkages between each BSC dimension or KPIs and HCW satisfaction and loyalty, as explained in the perspectives below. We made this choice since HCW satisfaction was deemed one of the latest affected perspectives in the strategic maps [53]. After that, we merged all the causal relationships into a single strategic map, which is shown here by Figure 4.

#### 1.5.1. Managerial Perspective

The role of health care management in improving HCW satisfaction and loyalty has been discussed in many studies [46]. Executives’ appreciation and recognition of HCWs’ efforts result in higher HCW satisfaction rates [31,42]. Other studies found that executives who have better communication and relationships with HCWs can better understand their needs and unfavorable working conditions. Consequently, this creates favorable working conditions [42,47]. Supervision is also critical to clarify job tasks and objectives and to cope with stress support [48,49]. A lack of roles and ambiguity increased HCWs’ dissatisfaction and lowered productivity and efficiency [50]. Additionally, a review found that most of the variance in intention to stay referred to managers respecting HCWs’ opinions [51]. The better the managers perform, the more doctors’ work engagement was also linked with higher doctors’ satisfaction [46].

#### 1.5.2. Financial Compensation and Rewards Perspective

Many studies have referred to financial compensation and motivations, including rewards, as another essential predictor. For example, many reviews [31,51,52] revealed that satisfaction with payment contributed to the greatest variance in job satisfaction. Financial compensation includes salary, incentives, and benefits packages [21,46]. Access to resources was also found to have a positive impact on doctors’ satisfaction [46].

#### 1.5.3. Knowledge and Growth Perspective

This perspective assesses HCWs’ competencies, knowledge, and skills development, as well as the training materials and HCWs’ accessibility to them [21,49]. A study found that on-the-job training motivated 99.0% of HCWs [22]. This perspective also measures the professional development opportunities of HCWs, such as promotion in their career [31,52]. Opportunity for professional development, being a chief, and prior achievement were found to have a positive impact on doctors’ satisfaction [46].

#### 1.5.4. Technology Perspective

In BSC, this perspective is usually combined with the information perspective. However, previous studies revealed the need to evaluate them separately [2,23]. The effect of technical and medical equipment on HCW satisfaction was assessed in this context [21]. It was found that an electronic decision support system could improve the work motivation of HCWs [49].

#### 1.5.5. Internal Process Perspective

This perspective contains the evaluation of job security [50]. Strategies to improve safety in the work environment could improve job satisfaction [21,47]. On the other hand, the lack of equipment or medication [24,49], such as the nonavailability of personal protective equipment during the pandemic, increased dissatisfaction [25]. Moreover, a high workload and HCW shortage negatively influenced HCWs’ job satisfaction [21,24,46,49].

#### 1.5.6. External Perspective

During the last two decades, both the social and environmental dimensions of sustainability have been gaining increasing attention among different stakeholder groups [6,26]. Social factors such as the community and patients’ appreciation [21], as well as the social status of the job [50] and organizational prestige [50], were found to increase HCWs’ job satisfaction. Moreover, family support was found to reduce burnout levels among HCWs [27], which in turn increases HCW satisfaction. On the other hand, other environmental factors, such as building-related factors and infrastructure, lighting, noise, and space, affected HCWs’ ability to work and consequently their satisfaction [21,49,50]. However, there is still limited research on the effect of these factors on customer loyalty, specifically HCW loyalty.

#### 1.5.7. Customer Perspective

Positive relationships and improved communication among staff and solidarity and teamwork among them improve their job satisfaction [21,46,47]. Moreover, work-life balance also positively affects HCWs’ job satisfaction [28]. On the other hand, emotional exhaustion is considered a symptom of HCWs’ burnout [29], and burnout is a predictor of job dissatisfaction [30].

HCW satisfaction is vital in the hospital quality process [42]. In the same vein, researchers highlighted that a job satisfaction survey should include key contextual factors affecting it [32]. On the other hand, a loyal attitude is a behavioral intention that reflects faithfulness to something [33]. HCW satisfaction can predict loyalty attitudes such as preference against competitors, recommendation willingness, and intention to stay or leave [34,35]. Intent to stay or leave was evaluated in studies that cannot measure turnover directly [36,46]. This is considered necessary since a lower turnover leads to lower recruitment and training costs, increased retention of valuable employees, and increased organizational commitment and loyalty [37,38]. Additionally, a study [38] revealed a negative relationship between job satisfaction and the intention of nurses to quit their current hospital.

In previous studies, validated items for loyalty measurement included satisfaction and loyalty attitude measurement, specifically recommendation and return intentions [34,39]. Work pride was a predictor of healthy working conditions [40]. We believe that work pride may affect HCW satisfaction and loyalty; however, this has not been assessed in the literature. Using a single item to assess actual patient satisfaction directly was suggested to be better than its assessment through multidimensional items [41,43,44].

#### 1.5.8. Sociodemographic Factors

In addition to the previously mentioned BSC perspectives, sociodemographic factors also impacted HCWs’ job satisfaction. Sociodemographic factors related to HCWs can be HCWs’ age [46], gender [46], profession type [24], specialty [46], marital status [24,46], years of work [46,52], and educational level [52]. Years of work were negatively associated with job satisfaction [52]. Additionally, educational level was found to have an inverse relationship with job satisfaction [52]. However, another study found that bachelor’s holders had higher job satisfaction than diploma holders [38], which could have been referred to as the more increased workload among diploma holders. On the other hand, organizational characteristics were also found to affect job satisfaction [45]. Additionally, it was found that hospital type and structure have a significant impact on physician satisfaction [46]. Administrative types can affect the hospital’s strategy, including its mission and vision, which may have an effect on the performance of BSC perspectives. However, the effect of hospitals’ administrative types on the previously mentioned factors has yet to be studied.

## 2. Methods

### 2.1. Research Design

This study is part of a broad project to use the BSC to strategically improve Palestinian hospitals through the analysis of their weaknesses and strengths based on BSC perspectives. This research is a cross-sectional study. The questionnaire was developed using Kaplan and Norton’s theoretical framework [1,2], and it was validated using the best methods for constructing and validating the health and behavioral scales [54].

### 2.2. Item and Scale Generation

BSC-HCW1 was developed using the previously reported technique for BSC-PATIENT development [23] with HCW adaptation. The same two panel experts conducted item and scale generation. The first panel examined the item face validity [55] per subdimension. This group then used a five-round Delphi technique [56]. Second, members of the second panel, consisting of 13 senior hospital executives from four Palestinian hospitals, were asked to assess the importance of 45 subdimensions to the strategic development of hospitals on a 10-point semantic scale. In addition, hospital executives were invited to indicate any additional important subdimension or KPI that was not included on the list. The characteristics and sociodemographics of this panel were described in a previous study [23]. For the following stage, we identified the important subdimensions. We specified an average score of seven as a threshold. These efforts were made in tandem with the creation of BSC-PATIENT [23]. As a consequence, 58 items remained. The second panel employed four- and three-point ordinal scales to score the relevance of each item [57]. The first author calculated the item content validity index (I-CVI), the scale content validity index (S-CVI), and the universal agreement among experts for the content validity index (CVI-UA) [57] to examine the content validity per item and scale. Items with a CVI score of less than 0.60 were removed. The items with a score of 0.6-0.8 were re-evaluated [57]. See Figure 5 below.

The panelists chose a three-point Likert scale: yes, neutral (I do not know), and no. Reasons for that were the high number of the remaining items, evidence of a faster and higher response rate on a three-point Likert scale than a five-point Likert scale [58], and the opportunity to check item availability through yes/no questions contributed to this decision. Furthermore, this scale was deemed more appropriate due to the pandemic’s impact on hospitals and HCWs [17,18,19]. Finally, all authors were requested to review the instrument, and the necessary changes were made.

### 2.3. Linguistic Validation and Translation

The same techniques used for linguistic validation and translation of BSC-PATIENT questionnaire items were used [23].

### 2.4. Pretest and Internal Consistency

Internal consistencies of the instrument’s perspectives in the initial edition of the questionnaire were evaluated. The first version of the questionnaire was pretested on 30 HCWs in one NGO hospital in the south of the West Bank. We asked them for their opinion on the language’s simplicity. We also kept track of how long it took them to complete the questionnaire. Items were assigned codes. Afterward, Cronbach’s alpha [59] was calculated using IBM SPSS statistics 21 software. Values greater than 0.6 were deemed appropriate for each perspective. As a consequence, few elements were changed or removed.

### 2.5. Sampling Procedure and Power Calculation

Since this is a part of broad research, the same sampling procedure and HCOs sample used to produce BSC-PATIENT [23] was also used to develop BSC-HCW1. Between June and December 2020, requests were sent to 15 hospitals on the West Bank and three hospitals in Jerusalem. Convenience sampling was used to choose the hospital sample. However, the total number of beds per administrative style and governorate were taken into account when selecting the participants (HCOs and HCWs).

Using the Steven K. Thompson sample size equation [60],
$$n=\frac{N\times p\left(1-p\right)}{\left[N-1\times \left(d^{2\;}÷z^{2}\right)\right]+p\left(1-p\right)}$$
where *n* is the sample size, *N* is the population size, *p* is the estimated population variability (0.5), *d* is the margin of error (0.05), and the *z* score is at the 95 percent confidence interval (1.96). In our research, *N* was the number of HCWs in Palestinian hospitals, which is 36,809 [61]. The required sample size was 381 HCW. In addition, researchers have recommended that 200 participants or five responders per parameter are appropriate sample sizes for exploratory factor analysis (EFA) [62,63,64]. To test structural validity, the sample is split to perform EFA and confirmatory factor analysis (CFA) [65]. The authors were concerned about the low response rate due to the pandemic impact on hospitals and HCWs’ high workload. Therefore, a total of 800 questionnaires were distributed.

### 2.6. Ethical Consideration

The research was approved by the Institutional Review Board (IRB) on 31 May 2020. The Research and Ethics Committee at An-Najah National University’s Faculty of Medicine and Health Sciences authorized all of the steps stated in this study with the reference code number (Mas, May/20/16).

First, we applied for the Palestinian Ministry of Health to obtain approval to perform the research in public hospitals. The request was then applied to each hospital separately in all administrative types. Between January and October 2021, the final form of the questionnaire was distributed. In addition, all HCWs were requested to sign a written consent form agreeing to participate. They were informed that their answers would be kept private from their management. Furthermore, all HCWs were advised that answering the questionnaire was entirely optional, and they might decline or withdraw at any moment.

### 2.7. Data Collection and Participants

The data were collected by the first author and four medical students from An-Najah University. The first author set three hours instructing each medical student on BSC, data collection processes, and ethics. The tasks and hospitals were delegated to the medical students based on their residence: eastern Jerusalem and the north, middle, and south of the West Bank. However, the Gaza Strip was omitted from the research due to political circumstances and accessibility issues. Furthermore, five hospitals were excluded: two nonoperating military hospitals, one mental health hospital, and two rehabilitation hospitals. For best representativeness, the maximum variation sampling approach was utilized. Hence, we looked for variance in our sample in terms of hospital size, location, and administrative type [66]. The HCWs were conveniently selected in this study. It was explained to them that participation was optional. Printed questionnaires were sent to respondents instead of sending them through e-mail [67].

Since experiences and attitudes might sometimes be unknown [68], the “I do not know (neutral)” response was introduced as an option to prevent response bias [67]. Second, the data collectors checked the questionnaires upon recovery to ensure that the number of missing responses was minimal. They brought the participant’s attention to missing pieces and asked them to fill them in. If any questions were still lacking upon entering data, they were recorded as I do not know. The inclusion and exclusion criteria were established as a Palestinian doctor or nurse of either gender who had worked at the examined hospital for at least three months. The included departments were emergency, internal medicine, surgery, gynecology, and pediatrics.

### 2.8. Statistical Analysis

The Shapiro–Wilk test was used to check the normality of the data. The frequencies were utilized to assess HCWs’ sociodemographics and the characteristics of the participating HCOs.

#### 2.8.1. EFA

EFA was conducted with the Promax rotation approach [69] to examine structural validity for 254 responses. To assess the adequacy of the EFA, the Kaiser–Meyer–Olkin (KMO) and Bartlett’s sphericity tests were used [70]. An eigenvalue higher than one [71] and a visual assessment of Cattell’s scree plot [71] were used to decide whether a component was included or excluded. A factor loading of 0.50 and greater than all cross-loadings of other constructs determined item inclusion or exclusion [63]. For this part, IBM SPSS statistics 21 software was used.

#### 2.8.2. CFA

The remaining 200 responses of the sample were subjected to CFA. The maximum likelihood estimation approach was used in IBM Amos 23 Graphics software (IBM, Wexford, PA, USA). The most often used fit indices were utilized to assess the goodness of fit of the competing models. The minimum discrepancies were split by degrees of freedom less than five and closer to zero, a *p* value greater than 0.05, the goodness-of-fit index (GFI), the comparative fit index (CFI), Tucker–index Lewis’s (TLI), with cutoff values near 0.95, root mean square error of approximation (RMSEA) of 0.06 and a standardized root mean square residual (SRMR) value of 0.08. [72,73]. The item inclusion-exclusion decision was set to be based on a factor loading higher than 0.50.

#### 2.8.3. Correlations

The interitem correlation (IIC) and corrected item-total correlation (CITC) were then computed [74]. Items with a correlation greater than 0.85 were considered redundant and eliminated in this analysis [75]. The bottom limit was set at a correlation of 0.30. In addition, the composite reliability (CR) per component was assessed to evaluate the internal consistency. CR is preferred over Cronbach’s alpha, specifically in structural equation modeling [76]. A CR of 0.60 was deemed adequate [77,78].

#### 2.8.4. Convergent and Discriminant/Divergent Validity

Finally, the Fornell-Lacker criterion [79] was employed to assess convergent and discriminant/divergent validity. If the computed average variance extracted (AVE) was more than 0.50, convergent validity was regarded as appropriate [80]. However, if a value of 0.50 was used with a CR greater than 0.60, the construct’s convergent validity was still regarded as satisfactory [79]. To prove discriminant validity, the AVE square root (SQRT) should be larger than the correlations with other latent components [77]. Furthermore, the factor’s uniqueness was assessed based on the value of Spearman correlation (r) with other factors at the same scale. As a result, we calculated r, which was classified as negligible when r < 0.20, low (r = 0.20–0.49), moderate (r = 0.50–0.69), high (r = 0.70–0.85), or very high (r = 0.86–1.00) [81,82]. The lack of a high or very high r between the subscale factors in this study indicated discriminant validity [82].

## 3. Results

### 3.1. Item Generation and Scoring

In the content validity assessment, the I-CVI results led to the removal of three items and indicated that 15 items required revision. The revised items necessitated additional explanation and rewording. This step increased the S-CVI and CVI-UA from 0.90 and 0.72 to 0.94 and 0.76, respectively.

### 3.2. The Instrument’s Structure and Items

The section of HCWs’ sociodemographics included age, gender, profession type, working department, years of experience, and total monthly income. Moreover, the questionnaires were coded based on the hospital name, administrative type, location, and JCI accreditation. The second section of the questionnaire was designed to evaluate HCW satisfaction predictors based on BSC perspectives and to directly measure their satisfaction and loyalty.

#### 3.2.1. The Managerial Perspective

This section included (a) an evaluation of managerial performance; (b) the relationship between management and HCWs, such as mutual respect, continuous communication, managerial support, delegation, engagement, authority, and recognition; (c) the managerial role in HCWs’ performance assessment; (d) the clarity of hospital strategy, including its mission and vision and its connection to work plans; and (e) the HCWs’ trust in their manager.

#### 3.2.2. The Financial Perspective

It contained five questions that asked the HCWs to evaluate their salary suitability for their competencies and responsibilities, performance-related financial motivations, compensation fairness, salary slip, and other financial packages and risk-related insurance premiums.

#### 3.2.3. The Internal Perspective

This section contained (a) two questions assessing the implementation of safety standards and the education the HCW received on infection control and safety standards; (b) five questions evaluating the time dimension, including the workload compatibility with the time given, the time spent with each patient, the resting time, and the work-life balance; and (c) three questions to evaluate the supplies and medication quality and the quality prioritization at the hospital in its provided services.

#### 3.2.4. The Knowledge and Growth Perspective

This section included (a) seven questions addressing the knowledge and growth perspective; (b) three questions that included guidelines on diseases, medication related to HCWs’ specialty, infection control, and safety measures; (c) two questions that assessed HCWs’ accessibility to knowledge and research, and research productivity motivations; and (d) two questions that were used to evaluate job description clarity and the introductory period.

#### 3.2.5. The Technology Perspective

It included six questions to evaluate the availability of a medical information system at the hospital and the training provided to HCWs to guide their use, the ease of use and the evaluation for this system in making accessibility to patient records and reports easier and faster and making HCWs work more productive and efficient.

#### 3.2.6. The Environment/External Perspective

This section assessed (a) the hospital location in reference to HCWs’ residency and the ease of access in emergency cases and (b) the hospital reputation compared to other hospitals.

#### 3.2.7. The Customer Perspective

This section assessed (a) internal customer factors: HCW satisfaction, intent to stay, recommending hospital to colleagues, teamwork, and emotional exhaustion; (b) external customer factors: the respect of patients toward HCWs was evaluated.

Finally, three items in the instrument were designed to be reversed in the statistical analysis: ESS1, which assessed the blame of HCWs when reporting medical errors. Additionally, ESB1 and ESB2 considered HCWs’ emotional exhaustion.

### 3.3. The Pretest and the Internal Consistency

The pretest was conducted in a nongovernmental hospital in the south of the West Bank. The questionnaire length was deemed to be adequate by HCWs. In addition, the design was well accepted and easy to understand. HCWs made specific small suggestions, which were taken into account. These suggestions were related to a few items that had been reworded. The questionnaire took approximately 7–10 min to complete.

After piloting, Cronbach’s alpha was calculated for each BSC perspective. The Cronbach’s alpha was 0.88, 0.63, 0.80, 0.54, 0.83, 0.88, and 0.81 for the managerial, financial, internal, external, knowledge and growth, technology, and customer perspectives, respectively. To raise Cronbach’s alpha, we decided to delete four items: ESF4 and ESF5 from the financial perspective and ESC1 and ESC3 from the customer perspective. We also decided to separate the reputation items from the accessibility items from the external perspective and move it to the customer perspective, which raised the Cronbach’s alpha to 0.72 and 0.83, respectively. In conclusion, 51 items remained. The Cronbach’s alpha for the instrument was 0.94.

### 3.4. Linguistic Validation and Translation

The final questionnaire forms in English and Arabic were completed and ready to be used.

### 3.5. Sample Size and Characteristics

Hospital approvals took six to nine months to obtain since the research took place during the COVID-19 pandemic. Only 15 of the 18 hospitals consented to participate. The data were collected between January and October 2021. The results of the hospital that was included in the pretest were excluded. Then, at the remaining 14 hospitals, we delivered 800 questionnaires, out of which 454 valid questionnaires were retrieved (response rate was 57%). Table 1 and Table 2 reveal the characteristics and sociodemographics of the respondents.

### 3.6. Statistical Analysis

#### 3.6.1. Testing the Normal Distribution

The data were not normally distributed. Therefore, nonparametric tests, specifically Spearman correlations, were chosen in the following steps.

#### 3.6.2. Structural Validity in EFA

EFA for the 51 items resulted in 35 item loadings higher than 0.50 for 15 components. All the components had eigenvalues greater than one. The KMO was 0.832 with a significant Bartlett’s test, indicating a high level of sample adequacy [77,83]. The total variation was 66.72%. See Table 3. The 15 components were technology (TECH), HCWs’ development (HCWDEV), management performance evaluation (MGMTEVAL), work-time and life balance (WTLB), loyalty (LOY), medical supplies and services quality (MSQUAL), financial incentives (FIN), HCWs’ engagement (ENG), reputation (REPUT), management communication (MGMT COMM), access (ACC), introductory period (ITRODP), safety (SAF), and no blame error reporting (NBR). However, no item had a loading higher than 0.5 on the 15th component. The scree plot results confirmed only 10 components out of 15, so these 10 were tested in the next step. To see the items that did not load in EFA or were suggested to be deleted based on Cattell’s scree plot assessment, check Appendix A.

#### 3.6.3. Structural Validity in CFA

CFA was performed for the resulting ten components in EFA. The CMIN/DF was 1.966. However, the other model fit indices were CFI = 0.885, GFI = 0.841, TLI = 0.860, RMSEA = 0.064, and SRMR = 0.0692, with a significant *p*-value. Hence, in the next phase, the model was tweaked based on the item loadings, model fit indices, and computations in the convergent, discriminant, CR, IIC, and CITC until the optimal model was reached. For example, the ESC4 item was removed from the MGMT COMM and was covered with a single item construct measuring managerial trust (MTR). Additionally, the REPUT component was converted to the patient with respect to HCWs (PTRs). ESR4 and ESL5 items were moved to the LOY construct. Moreover, items with loadings less than 0.5 were also removed or relocated to other constructs on which they had better loadings. Moreover, ESE2 and ESE3 items were added to the MGMTEVAL construct. Two constructs, MSQUAL and HCWDEV, were merged into one construct: quality and development (QUALDEV). This was due to the very high correlation between them. This merging also increased the fitness of the model. We did not transfer the items that were suggested to be removed based on Cattell’s scree plot except two; ESR4 and ESC5, which were added to the CFA model since we considered them important items and adding them did not lower the fit of the model in CFA. Finally, eight modification indices were utilized to improve the fit of the model. As a result, the optimal model consisted of nine constructs. The CMIN/DF was 1.334. Additionally, the other model fit indices were CFI = 0.958, GFI = 0.875, TLI = 0.948, RMSEA = 0.041, and SRMR = 0.0557. However, the *p*-value was significant. See Figure 6 and Table 4.

#### 3.6.4. Internal Consistency

The composite reliabilities for all factors were higher than 0.6. Additionally, all factors’ IIC and CTIC were higher than 0.3. The IIC ranged from 0.334–0.703, and the CITC ranged from 0.466–0.729, reflecting satisfactory internal consistency. See Table 5.

#### 3.6.5. Convergent and Discriminant Validity

For the five factors MGMTEVAL, ENG, QUALDEV, WTLB, and LOY, the convergent validity was between 0.4 and 0.5. However, the CRs for all were greater than 0.6, indicating acceptable convergent validity [79]. Correlations between the independent factors were insignificant or low in this context, except for the moderate association between the MGMTEVAL factor and ENG. No high or very high correlations were found between factors. On the other hand, the square roots of the AVE were higher than the off-diagonal correlations between components. In other words, convergent and discriminant validity were fulfilled for all factors, as seen in Table 5. See Table A1 in Appendix B for the final resulting items.

## 4. Discussion

### 4.1. Discussion of the Main Results

In line with this paper’s aim, we developed, translated, and validated the BSC-HCW1 instrument to engage HCWs in the evaluation process of BSC perspectives: the financial, internal, knowledge and growth, customer, external, and managerial perspectives. Our findings showed that the final model of BSC-HCW1 resulted in nine factors. Two factors represent dimensions from the managerial perspective: MGMTEVAL and MTR. The FIN factor represents a dimension of the financial perspective. The QUALDEV factor reflects a dimension of the internal process. The TECH factor refers to a dimension of the knowledge and growth perspective. Finally, four factors, ENG, WTLB, LOY, and PTR, represent dimensions from the customer perspective. MTR and PTR are single-item factors that are compatible with the recommendations of single-item use [41,43,44]. None of the designed variables from the external perspective, such as hospital accessibility and reputation, were loaded in our model. In general, the final BSC-HCW1 model demonstrated construct, convergent, and discriminant validity. P values were statistically significant in CFA because of its sensitivity to data normality. In addition, all of the CFA indices were higher than the cutoff limit, with the exception of the GFI, which was slightly lower than expected. However, according to a study, the GFI value may still be regarded as appropriate if it is more than 0.80 [85].

Additionally, the CR, IIC, and CITC were satisfactory. The occurrence of moderate correlations between factors might be attributed to the existence of causal links between BSC perspectives and dimensions, as numerous BSC studies [2,3] have suggested, not due to the lack of discriminant validity. Specifically, no high or very high correlations were found between factors. Therefore, the BSC-HCW1 proved to be a useful and valid tool to engage HCWs in a comprehensive assessment of BSC perspectives, financial, customer, internal process, knowledge and growth, and managerial.

The response rate was low, as expected by the authors, which was also perceived by other studies including HCWs during the same period [86,87]. This can be attributed to the high workload HCWs had during the pandemic. The response rate was lower among doctors, which is due to their higher workload and lower numbers than nurses in the Palestinian hospitals. This is compatible with two reviews [88,89] who found that the doctors’ response rate was lower than that for the general population and recommended effective methodologies to increase their response rate, such as financial incentives. Researchers have reported that a response rate of approximately 60% is accepted [90], which is very close to our response rate. We also distributed a much higher number of questionnaires than required as well as the well-adjusted representation of HCWs in our sample from several hospital types and various regions of Palestine, which suggests that our response rate given the current circumstances is acceptable, and the results are relevant and can be generalized. However, three constructs had fewer than three items, and two of them had a single item. In some cases, when a factor has a narrow scope and is unambiguous, using a single item to directly assess this variable is considered more favorable than using multidimensional items [41].

### 4.2. Comparison with BSC Studies

BSC reviews revealed that most of the previous implementations did not consider engaging HCWs in the BSC implementations [91,92,93,94,95]. The main focus was only on assessing the HCW satisfaction perspective without focusing on the other BSC perspectives [2]. Moreover, heterogeneity in the data collection tool used for evaluating HCW satisfaction was perceived [3,5]. This led to the inability to perform a meta-analysis for the BSC impact results [3].

From the 36 BSC implementations that resulted in the review of BSC dimensions [2], 69.44% did not include HCWs at all in the PE process. A total of 2.77% of the 36 implementations performed staff observations [94]. Only 22.22% of the implementations conducted interviews with HCWs [96,97,98,99,100,101,102,103], through which they evaluated the HCW satisfaction level. The use of the qualitative methodology was referred to due to the lack of prior evidence and inadequate existing theory [96]. However, two implementations distributed surveys to HCWs, which represented only 5.56% of BSC implementations [20,104]. One of them [104] asked a third party who benchmarks the hospital’s employee satisfaction against the other hospitals to measure their physician satisfaction and was presented as a sole KPI in the BSC evaluation, so the survey did not include engagement of HCWs in BSC perspectives evaluation. Another recent study [20] validated a survey to engage patients in BSC since they found that the number of tools to measure management practices of health facilities was very limited, and they could not find any evidence that the instruments designed for use in LMICs had been validated. Unlike the BSC-HCW1, the instrument KPIs were not designed based on a rigorous review of BSC perspectives and dimensions but were built based on the review of other managerial tools. Moreover, unlike the BSC-HCW1, the instrument was validated only using EFA analysis. The resulting dimensions were stakeholder engagement and communication, community-level activities, update of plan and target, performance management, staff attention to plan, target, and performance, and drugs and financial management. Therefore, the utilized dimensions mainly focused on evaluating to what extent HCWs engaged in management practices but did not actually engage HCWs in the process of PE from the BSC perspective. The authors of this instrument recommended that further investigation and refinement in this area is still worthy.

### 4.3. Strengths and Limitations

The BSC-HCW1 has several strengths. First, it is the first validated instrument designed to engage HCWs in a comprehensive assessment of BSC perspectives, financial, customer, internal process, knowledge and growth, and managerial. Second, this is the first validated instrument to conduct PE for Palestinian hospitals based on HCW opinions and observations. BSC-HCW1 will help the Palestinian Ministry of Health and health policy makers improve the performance of the health sector and overcome many challenges. For example, there is a lack of existing data measuring such KPIs in the records of many Palestinian hospitals. Additionally, there was a lack of transparency and the unwillingness of many hospitals to share the data extracted from their hospital records externally. The success in using BSC-HCW1 in the Palestinian health care context, which is characterized by fragility and fragmentation both geographically and administratively, may indicate that this instrument can be utilized successfully in other hospitals in LMICs or countries that reside under complex situations. Finally, the BSC-HCW1 will solve the heterogenicity in KPIs that was perceived in the previous BSC implementations and will offer a uniform assessment. This will facilitate PE comparisons among hospitals based on area and administrative style. It will also enhance data sharing among hospitals and recommendations among researchers, which will lead to improving hospital performance and a better understanding of HCW satisfaction and loyalty predictors worldwide.

On the other hand, this instrument has some limitations. First, the external perspective dimensions were ultimately excluded during the validation process. A refinment of these perpective items may include it in future versions of the BSC-HCW1. Second, this instrument is solely intended for use by two specific categories of HCWs: doctors and nurses. Both categories are important, as they spend the majority of their time with patients and are ultimately in charge of providing care. However, other categories of HCWs who work in hospitals, such as technicians, pharmacists, and nonclinical HCWs, were not included in this study. Therefore, future versions to include these categories can be beneficial. Third, despite the validation of this instrument during the pandemic, it was developed prior to it, so it lacks essential items. For example, the assessment of personal protective equipment’s availability at hospitals during the pandemic. It also lacks an assessment of customer-related variables in this era, such as HCWs’ stress and fear and items related to the development and knowledge pertaining to COVID-19 updates. Therefore, it is recommended to consider adding such items to future versions. Moreover, it is recommended to include items that measure types of burnout other than emotional exhaustion from the customer perspective. Additionally, it is advised to include family-related factors and marital status in the instrument since they may work as modifiers for HCW satisfaction and loyalty. Moreover, we recommend adding items that assess motivation, work control, work stability, access to resources, and prior achievements since they may be predictors for HCW satisfaction. Furthermore, some HCWs noted that they were hesitant to provide negative feedback regarding their management performance, which may have biased the responses. However, all respondents were informed of the consent form’s anonymity and privacy to lower this bias. Additionally, this was explained to them verbally by the data collectors. Additionally, participant bias may have occurred since the sample was convenient and the included hospitals agreed to participate in the research. Nevertheless, the high percentage of the included hospitals (30%) from the total number of hospitals at West Bank and including all administrative style types from all regions may have reduced the selection bias. Another limitation is that due to our inability to access English-speaking patients, we could not verify this instrument in English. Future studies should include the psychometric properties of the BSC-HCW1 in an English-speaking country. Last, because of the vast number of KPIs, the developers of this instrument have decided to only include those dimensions that are directly relevant to the demands of HCWs from each BSC perspective. The development of a second version of BSC-HCW1 that examines the unrelated dimensions to HCWs’ demands at each BSC perspective has the potential to significantly improve the level of HCWs’ engagement in the PE of their hospitals and BSC implementations.

### 4.4. Practical Implications

It is strongly recommended that HCO managers worldwide make use of the BSC-HCW1 instrument in future BSC deployments. Researchers need to validate the instrument in other languages and countries worldwide. Consequently, the managers of HCOs will first be able to identify the strengths and shortcomings in the BSC perspectives and dimensions based on the judgments of HCWs. Second, managers will be able to identify which BSC dimensions are predictors of HCW satisfaction and loyalty by invloving HCWs in the evaluation of strategic map dimensions. Eventually, this will provide managers with a direction on how to create their future action plans and where resources should be allocated. Therefore, instead of concentrating only on the level of satisfaction perceived by HCWs, the BSC-HCW1 may be used in the performance evaluation of HCOs in general to assess a number of other aspects. The in-depth analysis offered by this tool will make a contribution to the area of health management in general and to BSC implementations in particular.

On the other hand, some BSC implementations [104] utilized a third-party services outsourcing to benchmark the hospital’s HCW satisfaction against all the other hospitals, while using BSC-HCW1 will offer hospital managers an easy and inexpensive implementation to engage HCWs in the PE of hospitals. Based on our observation, the time required for a typical implementation depends on the cooperation of the HCWs. In our case, each hospital required an average month of data collection after receiving approval due to the high workload during the first period of COVID, which may have made it harder for us to accomplish the task. The other reason was that we also distributed the patient questionnaire during the same period at each hospital. In a typical situation, if the HCWs were cooperative, we expected that it would take only one week. However, a cross-sectional application for this instrument will only lead to a first or second generation of BSC. If the hospital intends to apply a third-generation BSC, then we recommend at least one year between the first and the last measurement to assess the impact of implementation as per the resulting implementations in our systematic review [3]. Additionally, monthly or quarterly targets, action plans, and periodic evaluations using BSC-HCW1 and follow-up are needed. HCO managers need to figure out how to motivate HCWs to participate in the process by offering financial motivations and sharing the final results with them, including how their evaluation participated in improving the PE of their hospital. Additionally, HCO managers should ensure HCWs that they will not impose any accountability for them based on their evaluations. The effect of using BSC-HCW may differ from one setting to another. This needs further investigation.

## 5. Conclusions

Researchers and hospital administrators who want to adopt BSC in hospitals may benefit from utilizing BSC-HCW1. This instrument might help understand the performance of the perspectives and dimensions of BSC based on the opinions and observations of HCWs. Most BSC implementation studies did not include HCWs at all or included them simply to gauge their level of satisfaction. Additionally, HCWs’ loyalty was rarely taken into account. None of the BSC implementations were able to encourage the HCW to participate in the process of evaluating the perspectives and dimensions of BSC. The BSC-HCW1 is the first instrument that has been designed specifically to include HCWs in the process of conducting PE using BSC perspectives and dimensions. BSC-HCW1 might let hospital managers look at BSC strategic maps based on what HCWs have observed and what they think. Therefore, it is strongly recommended that researchers make use of BSC-HCW1 in any future BSC implementation. Another study is needed to produce another instrument that engages HCWs in evaluating the BSC dimensions that are not directly relevant to their needs but are nonetheless related to the PE of HCOs. It is essential that, in addition to HCWs, other stakeholders, such as patients and hospital administrators, be included in the implementation of BSCs. Palestinian health policymakers and hospital management will be able to assess their strengths and shortcomings based on the observations and views of their HCWs using this instrument. It is possible to make use of this validated instrument in its Arabic form in other Arab nations. However, validation in more languages is still required for this instrument.

## Figures and Tables

**Figure 1 ijerph-19-09096-f001:**
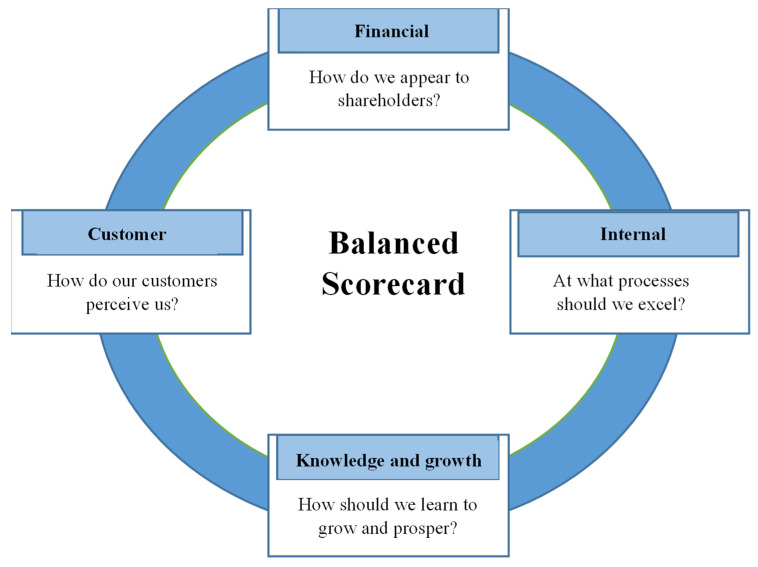
Balanced scorecard perspectives [1].

**Figure 2 ijerph-19-09096-f002:**
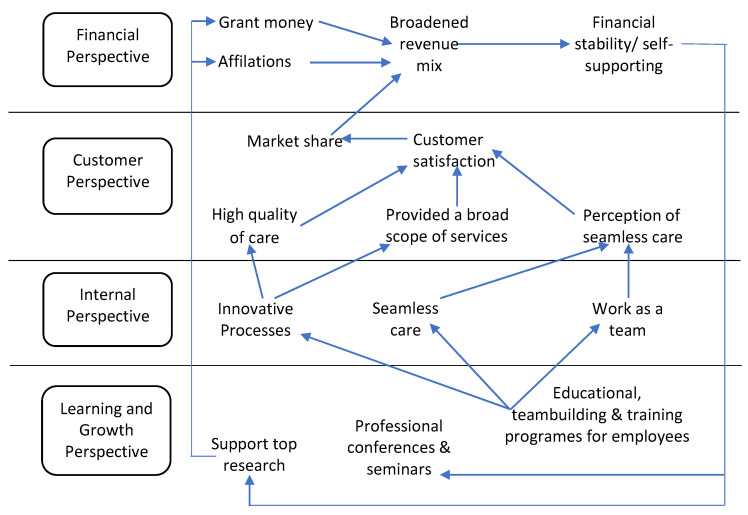
Duke University health system strategic map [7].

**Figure 3 ijerph-19-09096-f003:**
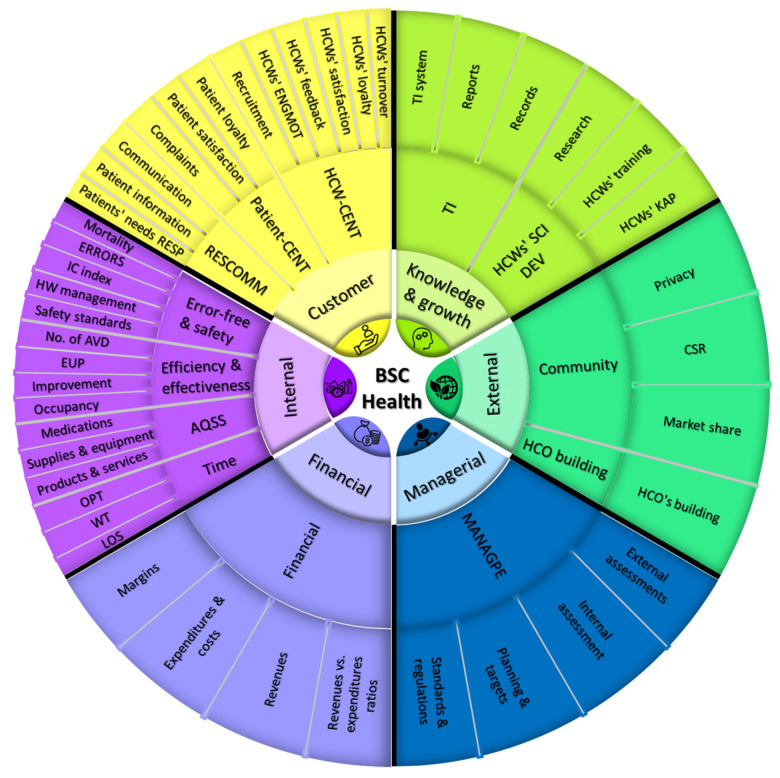
A summary of BSC perspectives in health care and their contents [2]. Figure legend suggests: Summary of BSC perspectives and the underlying major and minor subdimensions for the PE of HCOs. Note: BSC, balanced scorecard; HCWs, health care workers; HCOs, health care organizations; IC, infection control; HW, health waste; WT, waiting time; LOS, length of stay; KAP knowledge, attitude, and practices; TI, technology and information; CSR, corporate social responsibility; ERRORS, errors, accidents, and complications; No. of AVD, number of admissions, visits, and diseases; EUP, efficiency, utilization, and productivity; AQSS, availability and quality of supplies and services; OPT, operation processing time; RESCOMM, response to patients’ needs; Patient-CENT, patient-centeredness; ENGMOT, HCWs’ engagement and motivation; HCW-CENT, HCW-centeredness; MANAGPE, managerial tasks and performance evaluation; SCIDEV, scientific development.

**Figure 4 ijerph-19-09096-f004:**
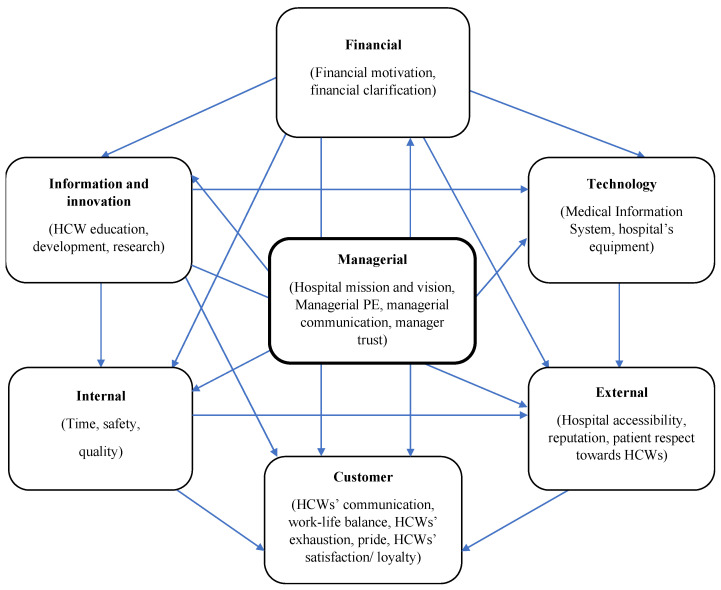
The conceptual model for the strategic map of the BSC-HCW1. Note: HCW, health care worker; PE, performance evaluation.

**Figure 5 ijerph-19-09096-f005:**
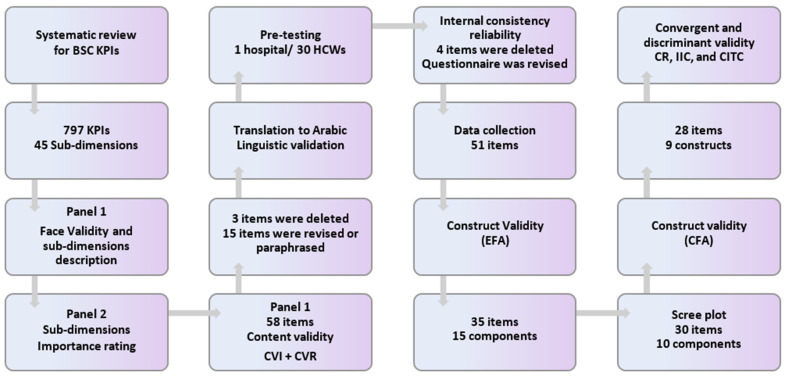
Flow chart of the development and validation process of the BSC-HCW1 instrument. Note: BSC KPI, balanced scorecard key performance indicators; CVI, content validity index; CVR, content validity ratio; HCW, health care worker; CR, composite reliability; IIC, interitem correlation; CITC, corrected item-total correlation; EFA, exploratory factor analysis; CFA, confirmatory factor analysis.

**Figure 6 ijerph-19-09096-f006:**
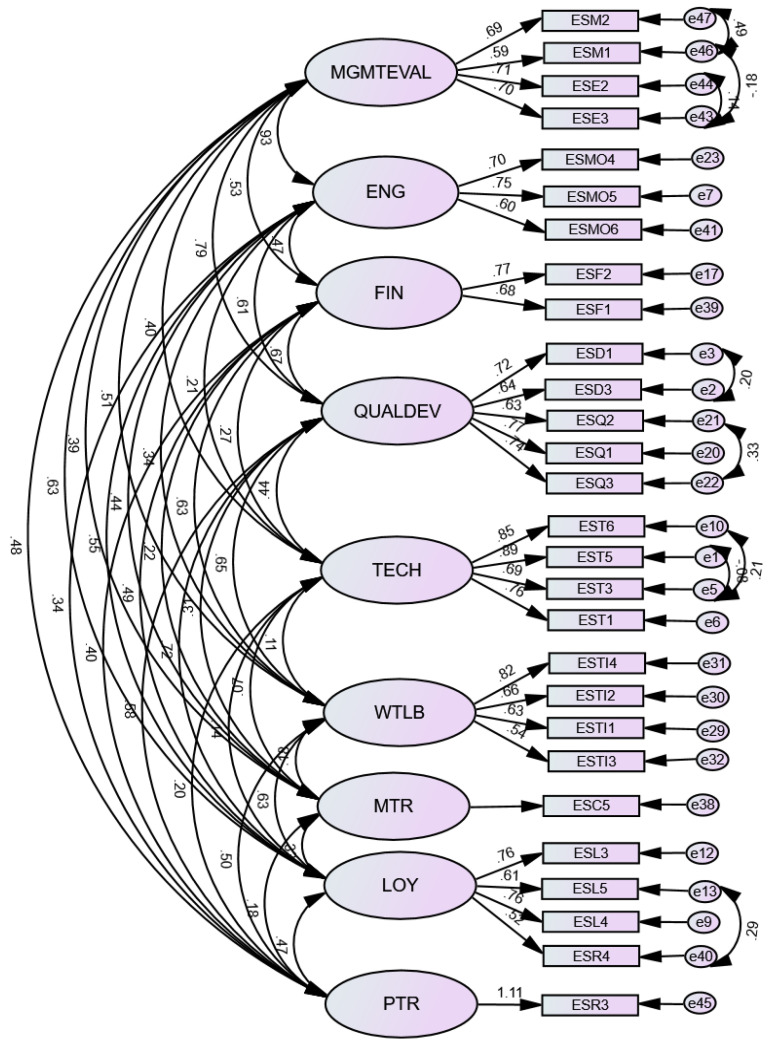
Confirmatory factor analysis (CFA) for BSC-HCW1 constructs. Note: MGMTEVAL, management performance evaluation; ENG, health care workers’ engagement; FIN, financial incentives; QUALDEV, quality and development; TECH, technology; WTLB, work time–life balance; LOY, loyalty; MTR, managerial trust; PTR, patient respect toward health care workers.

**Table 1 ijerph-19-09096-t001:** Sociodemographic characteristics of respondents (HCWs), N = 454.

Characteristics	*n*	%	Characteristics	*n*	%
Age			Years of experience		
20–29 years	198	43.6	0–2 years	149	32.8
30–39 years	163	35.9	3–5 years	107	23.6
40–49 years	59	13	6–9 years	79	17.4
50–59 years	26	5.7	10–13 years	45	9.9
60 years or above	8	1.8	More than 14 years	74	16.3
Gender			Income		
Female	232	51.1	2000–3000 NIS	108	23.8
Male	222	48.9	3001–4000 NIS	129	28.4
Department			4001–5000 NIS	101	22.2
Mixed	18	4.0	5001–6000 NIS	50	11.0
Pediatric	73	16.1	Higher than 6000 NIS	66	14.5
Internal medicine	81	17.8	Profession		
Surgery	98	21.6	Doctor	156	34.4
Emergency	91	20.0	Nurse	298	65.6
Gynecology	93	20.5			

Note: NIS, New Israeli Shekel; UNRWA, The United Nations Relief and Works Agency for Palestine Refugees in the Near East; NGO, non-governmental organization; mixed, only in one hospital were the nurses not specified to work in one department and were rotated between different departments.

**Table 2 ijerph-19-09096-t002:** Number of HCWs and hospitals based on hospital characteristics.

	Number of HCWs (Total = 454)	%	Number of Hospitals (Total = 14)	%
Administrative style				
NGO	170	37	5	35.71
Public	145	32	5	35.71
Private	111	24	3	21.43
UNRWA	28	6	1	7.14
City				
Hebron	87	19.16	3	21.43
Jerusalem	40	8.81	1	7.14
Nablus	166	36.56	5	35.71
Qalqilya	28	6.17	1	7.14
Ramallah	92	20.26	3	21.43
Tulkarm	41	9.03	1	7.14
Area				
North	235	51.76	7	50.00
Middle	132	29.07	4	28.57
South	87	19.16	3	21.43
Accreditation status				
Yes	97	21.37	3	21.43
No	357	78.63	11	78.57
Size				
Small (No. of beds < 80)	133	29.30	5	35.71
Medium (No. of beds 80–160)	188	41.41	5	35.71
Large (No. of beds > 160)	133	29.30	4	28.57

Note: UNRWA, The United Nations Relief and Works Agency for Palestine Refugees in the Near East; NGO, non-governmental organization.

**Table 3 ijerph-19-09096-t003:** Exploratory factor analysis (EFA).

Component	Item	Item Code	Factor														
			1	2	3	4	5	6	7	8	9	10	11	12	13	14	15
TECH	Hospital information systems and technology make access to patients’ records easier, faster, and more accurate.	EST4	0.923														
	The hospital information system and technology make generating reports easier, faster, and more accurate.	EST5	0.863														
	The hospital information system and technology make my work efficient and productive.	EST6	0.767														
	I believe that the hospital information system interface is user-friendly.	EST3	0.736														
HCWDEV	The hospital provides me with education on medication updates related to my specialty.	ESD3		0.972													
	The hospital provides me with access to the latest medical books and journals.	ESD4		0.811													
	The hospital provides me with educational updates regarding the diseases in my specialty.	ESD1		0.721													
	This hospital provides me with an access to the newest books, databases, and scientific papers.	ESD5		0.705													
MGMTEVAL	I believe that my superiors have the required competencies for their positions.	ESM1			0.951												
	My superiors are making the right decisions in work which support the hospital strategy.	ESM2			0.804												
	The management in this hospital asks for staff feedback, perceptions, and care for their satisfaction.	ESM3			0.515												
WTLB	The quantity of work assigned to me is reasonable with the time given.	ESTI4				0.708											
	I have sufficient time to rest and eat during my working day.	ESTI1				0.668											
	I can make a work–life balance and good time management.	ESTI3				0.660											
	I can spend sufficient time with each patient.	ESTI2				0.596											
LOY	My overall satisfaction is high.	ESL4					0.627										
	I want to keep working in this hospital for several years.	ESL3					0.599										
MSQUAL	The hospital medications and disposables are of high quality.	ESQ2						0.939									
	The hospital equipment helps me in offering high-quality medical services for patients.	ESQ1						0.685									
	Quality is a top priority at this hospital.	ESQ3						0.587									
FIN	I receive financial incentives based on my performance.	ESF2							0.836								
	I feel that my salary suits my responsibilities and competencies.	ESF1							0.529								
ENG	My manager engages me in the planning and decision-making process.	ESMO5								0.670							
	My manager understands and adequately supports me when I face an urgent, complex situation.	ESMO4								0.604							
	I am given enough authority and power to make decisions in my position.	ESMO6								0.536							
REPUT	I am proud to work with this hospital.	ESR4									0.653						
	I believe that patients respect health care workers at this hospital and trust them.	ESR2									0.637						
	I believe that this hospital has a better reputation than other hospitals in Palestinian.	ESR3									0.533						
MGMT COMM	Communication with management is frequent, and they keep me updated with sufficient information to do my job.	ESC4										0.839					
	I trust what my direct manager tells me or promises me.	ESC5										0.651					
ACC	It is easy to access the hospital when a case is urgent.	ESA2											0.937				
	Hospital location is close to where I live.	ESA1											0.672				
ITRODP	New employees are well introduced to the job description, and the specifications are clear in the job contract.	ESEM1												0.615			
SAF	Safety standards are implemented and assured (masks, gloves, sanitizers, etc.).	ESS2													0.663		
NBR	When errors are reported a blame free policy is taken by managers.	ESS1														0.503	
Percentage of variance (%) Total variance = 66.72%			21.08	7.25	4.80	4.46	3.87	3.55	2.86	2.71	2.62	2.49	2.39	2.34	2.16	2.07	2.07
Eigenvalues			10.75	3.70	2.45	2.27	1.97	1.81	1.46	1.38	1.33	1.27	1.22	1.19	1.10	1.06	1.05

Note: TECH, technology; HCWDEV, health care worker development; MGMTEVAL, management performance evaluation; WTLB, work time–life balance; LOY, loyalty; MSQUAL, medical supplies and services quality; FIN, financial incentives; ENG, HCW engagement; REPUT, reputation; MGMT COMM, management communication; ACC, accessibility; ITRODP, introductory period; SAF, safety; NBR, no blame error reporting.

**Table 4 ijerph-19-09096-t004:** The goodness-of-fit indices in EFA and CFA and results.

EFA [77,80,83]		CFA [84]	
Criteria for Good Fit	Measurements	Criteria for Good Fit	Measurements
- KMO: 0.6: low adequacy 0.7: medium adequacy 0.8: high adequacy 0.9: very high adequacy - Bartlett’s test *p*-value < 0.05 - Inclusion/exclusion criteria for the components: 1 - Eigenvalues ≥ 1 2 - Visual assessment of Cattell’s scree plot. -Inclusion/exclusion criteria for the items: 3 - The factor loading ≥ 0.50. 4 - Factor loadings on the assigned construct ≥ all cross-loading of other constructs.	- KMO = 0.832 (Chi square = 5442.68, degrees of freedom = 1275) - Bartlett’s test *p*-value < 0.001 - 15 components which have Eigenvalues above 1 - Cumulative variance = 66.72% - Cattell’s scree plot: keep 10 components	- χ^2^/df < 5 and closer to zero - The p-value > 0.05 - GFI - CFI - TLI GFI, CFI, and TLI close to 0.95 - RMSEA < 0.06 - SRMR ≤ 0.08	χ^2^/df = 1.33 *p*-value < 0.001 GFI = 0.875 CFI = 0.958 TLI = 0.948 RMSEA = 0.041 SRMR = 0.0557 - 9 constructs

Note: EFA, exploratory factor analysis; CFA, confirmatory factor analysis; KMO, Kaiser–Meyer–Olkin; χ^2^/df, minimum discrepancy divided by its degrees of freedom; GFI, the goodness-of-fit index; CFI, comparative fit index; TLI, Tucker–Lewis index; RMSEA, root mean square error of approximation; SRMR, standardized root mean square residual.

**Table 5 ijerph-19-09096-t005:** Factors IIC, CTIC, CR, convergent, and discriminant/divergent validity.

Factor	CR	IIC	CITC	AVE	MGMTEVAL	ENG	FIN	QUALDEV	TECH	WTLB	LOY	MTR
MGMTEVAL	0.769	0.373–0.701	0.550–0.653	0.455	**0.675**							
ENG	0.727	0.398–0.467	0.503–0.554	0.472	*0.503* **	**0.687**						
FIN	0.694	0.493	0.493	0.533	*0.288* ****	*0.216* ****	**0.730**					
QUALDEV	0.829	0.334–0.581	0.534–0.600	0.494	*0.492* ****	*0.364* ****	*0.392* ****	**0.702**				
TECH	0.878	0.483–0.703	0.620–0.729	0.645	*0.278* ****	*0.253* ****	*0.055*	*0.296* ****	**0.803**			
WTLB	0.760	0.345–0.484	0.483–0.610	0.448	*0.308* ****	*0.207* ****	*0.429* ****	*0.446* ****	*0.055*	**0.670**		
LOY	0.761	0.364–0.561	0.466–0.645	0.449	*0.407* ****	*0.310* ****	*0.341* ****	*0.476* ****	*0.209* ****	*0.455* ****	**0.670**	
MTR	-	-	-	-	*0.378* ****	*0.397* ****	*0.176* ****	*0.274* ****	*0.117* ***	*0.171* ****	*0.312* ****	-
PTR	-	-	-	-	*0.358* ****	*0.208* ****	*0.319* ****	*0.460* ****	*0.176* ****	*0.378* ****	*0.393* ****	*0.190* ****

Note: MGMTEVAL, management performance evaluation; ENG, health care workers’ engagement; FIN, financial incentives; QUALDEV, quality and development; TECH, technology; WTLB, work time–life balance; LOY, loyalty; MTR, managerial trust; PTR, patient respect toward health care workers; IIC, interitem correlation; CITC, corrected item total correlation; CR, composite reliability; AVE, average variance extracted calculated by the average square of loadings at each factor and used to evaluate the convergent validity; **Bold**, square roots of the average variance extracted; *Italic*, Spearman correlations between independent factors, both are used to assess discriminant validity; * *p* < 0.05; ** *p* < 0.01, -; single-item factor.

## Data Availability

The datasets generated and/or analyzed during the current study are not publicly available because the data are still not fully analyzed and the research is still in process but are available from the corresponding author (F.A.) upon reasonable request with the permission of the UNRWA, Palestinian Ministry of Health, and Al Makassed Hospital.

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
