# Peer review of "How to Engage Health Care Workers in the Evaluation of Hospitals: Development and Validation of BSC-HCW1—A Cross-Sectional Study"

_ijerph, 2022, doi:10.3390/ijerph19159096_

Round 1

Reviewer 1 Report

I believe that the study is very relevant and extremely promising. The methodology and results are well written. But the practical significance is not sufficiently presented. Medical institutions that would like to apply the proposed method should know 1) what is its laboriousness? 2) how much will it cost? 3) what is the time required for a typical implementation? 4) who can be attracted to conduct research (outsourcing), is it impossible for them to do it on their own? 5) what is the estimated effect that can be obtained (forecast)? Perhaps the answers to the questions posed will make the study better.

Author Response

We want to thank the reviewer for their comments and their evaluation to improve the manuscript.
Kindly find the attached file which contains a reply to your recommendations point by point.

Regards

Reviewer 2 Report

The current study presents interesting and important topic. The authors have followed a well thought out methodology to conduct the study and analyze the data. The paper aims to provide a tool that engages health workers in the implementation of  the balanced scorecard (BSC).

Regarding the paper, in my opinion few points need to be clarified:

·         Fig 1. and Fig.2 – Do the authors have the permission to use these schemes? Please explain.

·   According to the Authors: “The HCWs were conveniently selected based on their willingness to participate in this study (lines 359-3610). Could you please clarify, what does mean: “based on their willingness”. Please discuss also this information in the part: Limitations of the study.

·      In the discussion part (point 4.4. practical implications) please elaborate the international context of your study.

·    Additional references will be useful for the content of the study:  

1)      Other international research also reported that questionnaire research among medical workers have low response rates if compared with the general population:

·         VanGeest, J.B.; Johnson, T.P.; Welch, V.L. Methodologies for improving response rates in surveys of physicians: A systematic review. Eval. Health Prof. 2007, 30, 303–321.

·         Flanigan, T.S.; McFarlane, E.; Cook, S. Conducting survey research among physicians and other medical professionals: A review of current literature. ASA Proc. Sect. Surv. Res. Methods 2008, 36, 4136–4147

2) There is systematic review on factors associated with satisfaction of physicians employed in hospitals. Results of this systematic review could be useful for the analysis presented in point 6.1.

·    DomagaÅ‚a A, BaÅ‚a MM, Storman D, Peña-Sánchez JN, Åšwierz MJ, Kaczmarczyk M, Storman M. Factors Associated with Satisfaction of Hospital Physicians: A Systematic Review on European Data. International Journal of Environmental Research and Public Health. 2018; 15(11):2546. https://doi.org/10.3390/ijerph15112546

Author Response

(The authors gave the same response as above.)

Reviewer 3 Report

If possible to synthesize a little bit the methodology.

As the authors say that the number of the replies to the questionnaire is low, they can motivate why they consider the result relevant.

How the use of BSC can have an effect of motivation for HC workers?

Author Response

(The authors gave the same response as above.)
